# Glucocorticoid Receptor Blockers Pretreatment Did Not Improve Infarct Volume in Type-2 Diabetic Mouse Model of Stroke

**Rashmi Kumari [1],* and Lisa Willing [2]**

[1] Department of Neural & Behavioral Sciences, College of Medicine 500 University Drive, Hershey Medical Center, Hershey, PA 17033, USA

[2] Department of Neural and Behavioral Sciences, College of Medicine, Pennsylvania State University, Hershey, PA 17033, USA

* Correspondence: rkumari@pennstatehealth.psu.edu; Tel.: +1-717-531-5762; Fax: +1-717-531-5184

**Abstract:** Impaired glucocorticoid signaling in diabetes mellitus and its relation to suppressed immune function and hyperglycemia during acute stroke has been shown to be detrimental. Therefore, the aim of this study was to examine the effect of glucocorticoid receptor (GCR) blockers in a type-2 diabetic mouse model following hypoxia–ischemia (HI). We induced stroke in diabetic *db/db* and non-diabetic *db/+* mice by unilateral common carotid artery ligation followed by 20 min of HI. Mice were pretreated with RU-486, GCRII blocker (40 mg/kg), intraperitoneally, the day before, during stroke and post-HI. Blood and brain samples were collected at 24 h post-HI to measure blood glucose, corticosterone and infarct size. Similarly, another set of mice was pretreated with RU-486 + spironolactone, GCR1 blocker (25 mg/kg) subcutaneously for a week before inducing stroke and during recovery. Samples were collected at 48 h post-HI for various analyses. RU-486 treatment did not lower the blood glucose significantly, but RU-486 + spironolactone decreased the blood glucose in *db/db* mice post-HI. However, none of the treatment groups decreased the ischemia-induced serum corticosterone level or infarct size. This study suggests that even though GCR blockers improve hyperglycemia, they did not improve the infarct volume.

**Keywords:** glucocorticoid receptor; type-2 *db/db*; HI; corticosterone; hyperglycemia

## 1. Introduction

The dysregulation of hypothalamic-pituitary adrenal axis (HPA) and sympathetic nervous system has been reported both in preclinical and clinical studies of stroke [1,2]. The increased plasma corticosterone in rodents and cortisol in humans have been associated with high mortality rate and poor functional recovery in stroke patients [3–5]. A reason for poor stroke outcome was linked to post-stroke immunodepression and lymphocytopenia, and the blockade of the glucocorticoid receptor (GCR) prevented the post-ischemic lymphocytopenia [6]. Glucocorticoids are the ultimate mediator of the body's response to stress, and in conjunction with insulin and leptin, they exert a profound effect on regulation of energy consumption and expenditure [7]. Another function of glucocorticoids is to stimulate hepatic gluconeogenesis and reduce the ability of insulin to inhibit glucose production, which causes the increased plasma glucose in type-2 diabetes, triggering insulin resistance and central obesity [8,9]. In addition, elevated glucocorticoids have shown an impact on the development in *ob/ob* and *db/db* mice, where adrenalectomy prior to weaning prevents the development of obesity [10,11].

Glucocorticoid-induced hyperglycemia is common in both diabetic and non-diabetic patients. A previous study in an animal model of stroke suggested that a hyperglycemia-induced release of glucocorticoids worsened stroke outcome in a rat model of cardiac-arrest-induced transient global cerebral ischemia [12,13], whereas, in a clinical study, both

low and high cortisol levels were associated with increased mortality after stroke [5]. Other reports, by Sapolsky and colleagues, revealed that cells of the hippocampus expressing high levels of glucocorticoid receptors I and II were vulnerable to cell death [14]. Recently, an inhibitor of serum- glucocorticoid inducible kinase I (SGK1) was found to be protective in a type-1 diabetic mouse model of stroke [15]. Likewise, Kim et al. demonstrated that diabetes exacerbated stroke outcome which was associated with activation of the HPA axis and, inhibition of glucocorticoid synthesis, decreases infarct size via reducing IL-6 inflammatory response [1]. However, none of the studies investigated whether glucocorticoids are the main cause of ischemic damage, particularly in the type-2 diabetic mouse model of stroke. Based on this information, we planned to examine the effects of glucocorticoid receptor blockers I and II in a type-2 *db/db* mouse model of stroke.

Previously glucocorticoids have been used to treat edema associated with stroke, bacterial meningitis, and multiple sclerosis. However, the efficacy of glucocorticoid treatment was found to be both beneficial and detrimental in animal models of stroke [16,17]. In our clinical study, we observed increased hyperglycemia and acute immune suppression in diabetic patients up until 96 h post-stroke in addition to significant mortality and poorer recovery compared with non-diabetic stroke patients [18]. Similarly, we found a significant immune suppression in CNS in type-2 *db/db* and *ob/ob* mouse models following stroke [19,20]. Earlier reports indicated that increased corticosterone production in type 2 diabetes *db/db, ob/ob* mice and zucker (fa/fa) rats induced insulin resistance and increased glucose intolerance [21–23] and in a separate study, treatment of *ob/ob* mice with RU-486, a GCRII blocker, resulted in euglycemia and lowered circulating insulin levels [24]. Therefore, in this study, we treated the mice with glucocorticoid receptor (GCR) blockers before inducing stroke, and during recovery, to normalize hyperglycemia and improve the immune function, which in turn may reduce infarct size and improve stroke outcome in *db/db* mice compared with nontreatment control. Simultaneously, we compared the effect of GCR II blocker to the combined effect of GCR I and II blocker treatment in acute stroke.

## 2. Materials and Methods

### 2.1. Mouse Model of Hypoxia/Ischemia

All the experimental procedures in this study were approved by the Institutional Animal Care and Use Committee of Penn State University. Male type-2 diabetic *db/db* and their heterozygous nondiabetic control *db/+* mice were obtained from Jackson laboratory (JAX stock #004456) at 7–8 weeks of age. All mice were housed (2 per cage) in ventilated cages, maintained on a 12 h dark-light cycle with free access to food and water. After acclimatization, the mice were weighed, and blood glucose levels were measured with the BD Logic TM Meter (BD Pharmingen, San Jose, CA, USA) using a tail prick before treatment, after treatment and post-HI recovery. At 9–10 weeks of age, following the pre-treatment of RU-486 or combination of RU-486 + spiranolactone or vehicle, mice were exposed to hypoxia/ischemia as described previously with slight modification [19,25]. Mice were anesthetized with isoflurane (4% in 70% nitrous oxide/30% oxygen). A small midline neck incision was made, and the right common carotid artery was exposed and ligated twice using 3-O surgical silk and the animals were returned to their cages for 3 h of recovery during which time they had free access to food and water. Following recovery, hypoxia was induced by placing the mice in 500 mL jars (1 per jar) that were partially submerged in a circulating water bath maintained at 35.5 °C. The animals were exposed to a humidified gas mixture of 8% $O_2$ balanced with $N_2$ that was uniformly delivered to each jar for 20 min. The animals were returned to their cages and given free access to food and water for 24 and 48 h post-HI. Blood samples were collected at baseline prior to treatment, after treatment (day before the procedure) and at post-HI (24 or 48 h) for the measurement of corticosterone. At the time of sacrifice, mice were transcardially perfused/flushed with 30 mL 1× PBS. The brains were removed and frozen in isopentane at −30 °C and then stored at −80 °C prior to histological analysis

## 2.2. Glucocorticoid Receptor Blocker Treatment

In the first study, we tested the acute effect of glucocorticoid receptor blocker II to determine whether it can reduce stress-induced hyperglycemia and consequently improve stroke outcome. Both *db/db* and *db/+* mice were randomly separated into two groups and GCRII blocker, mifepristone (RU-486) or vehicle were injected intraperitoneally 24 h prior to surgery, a second dose was administered just after surgery, and a third dose 7 h post-HI. The non-treatment group received an equivalent volume of vehicle. RU-486 was obtained from (Sigma Aldrich, Burlington, MA, USA, Cat no. M8046) and dissolved in 70% ethanol/phosphate-buffered saline. As an initial acute study, the measurements were carried out at baseline (prior to treatment) and 24 h post-HI (see the study design Table 1).

**Table 1.** Study design.

| Treatment Stages | Study 1 (RU-486, 40 mg/kg; I.P.) | Study 2 (RU-486+ Spironolactone, 25 mg/kg; S.C.) |
|---|---|---|
| First dose | 24 h prior to surgery | daily injection 1 week prior to surgery |
| Second dose | just after surgery, | just after surgery |
| Third dose | 7 h post-HI, | 24 post-HI |
| Sample Collection | 24 h post-HI | 48 h post-HI |

The results of our initial study prompted us to examine whether treatment with a combination of glucocorticoid receptor blockers for a week prior to inducing stroke would lower the hyperglycemia and reduce the impact of stroke. In the second study, both *db/d* & *db/+* mice were divided into two groups. The treatment group received the combination of GCRII blocker, RU-486 (25 mg/kg) and GCR1 blocker, spironolactone, (25 mg/kg, Sigma Aldrich, USA, Cat no. S3378), dissolved in polyethylene glycol and non-treatment control received an equivalent volume of vehicle. The drug was injected subcutaneously daily, one week prior to the surgery until 48 h post-HI. The measurements were carried out prior to treatment (baseline), after a week treatment (after tt) and 48 h post-HI.

## 2.3. Histological Analysis of Stroke

Fresh frozen coronal brain sections (16 μm) were obtained by cryosectioning at regular intervals from striatum to hippocampus for hematoxylin and eosin (H & E) staining. H & E staining was performed as described previously [26]. A striatal and hippocampal section from each mouse brain was stored at −80 °C until staining. The fresh frozen brain sections were rehydrated using 2 min changes of ethanol (100%, 95%, and 70%). Then, sections were fixed in 70% ethanol for 5 min, rinsed twice in distilled water, stained with hematoxylin for 1.5 min, and rinsed in tap water for 10 min. The sections were dipped in eosin for 2 sec and rinsed in 70% ethanol. Then, sections were dehydrated with ethanol (70%, 95%, and 100%), cleared with xylene and cover slipped with Cytoseal[60] TM (Thermo Scientific). Each brain region was graded on an arbitrary scale of 0 to 7 under a microscope (Olympus BX-50). The grading was blinded to the treatment, section A (frontal brain: caudate putamen) scored cortex 0–1, caudate 0–2. Section B (midbrain/hippocampus) scored cortex 0–1, caudate 0–0.5, thalamus 0–1.5, hippocampus/dentate gyrus 0–1. The potential total of section A = 3 and B = 4 with 0 being no damage and A + B = 7 being the greatest infarct size.

## 2.4. Corticosterone Measurement

Retroorbital blood samples (50 μL) were collected under anesthesia at baseline, after treatment and specific intervals of recovery times. The samples were collected routinely around 10 AM in each experiment to prevent diurnal rhythm effects on corticosterone level. Serum was obtained by centrifugation at $16,000 \times g$ for 10 mins at room temperature and stored at −80 °C until the analysis. Serum corticosterone (CORT) was measured by radioimmunoassay using ImmuChem Double Antibody Corticosterone [125]I RIA Kit

(MP Biomedicals, LLC Diagnostic Division, Orangeburg, NY, USA), according to the manufacturer's instructions. Corticosterone values were calculated from a standard curve, which was generated using 0 to 1000 ng/mL CORT standards.

### 2.5. Statistical Analysis

The results are expressed as Mean ± Sem. Two-way ANOVA followed by Tukey's multiple comparison were applied to compare more than two groups and more than one variable to test the interaction between control and treatment at different time points for body weight, blood glucose, and corticosterone measurement. One-way ANOVA followed by Tukey's multiple comparison was used for stroke grading. The results were analyzed by Graph Pad Prism 6.0 and statistical significance was set at $p < 0.05$.

## 3. Results

### 3.1. Effect of Glucocorticoid Receptor Blockers on Body Weight and Blood Glucose

The body weight was significantly higher in *db/db* mice compared with their *db/+* vehicle and *db/+* treatment group at baseline. We did not observe a change in body weight at 24 h post-HI in either group as shown in Figure 1A. The pretreatment of RU-486 significantly decreased the blood glucose in the *db/db* group from baseline to 24 h post-HI (475.13 ± 26.76 vs. 253.78 ± 34.52 mg/dl). Similarly, the non-treatment *db/db* group also showed a decrease in blood glucose from baseline (556.31 ± 14.39 vs. 330.63 ± 65.76 mg/dl) as shown in Figure 1B. However, we did not find a significant effect of RU-486 on blood glucose between treatment and non-treatment at 24 h post-HI in *db/db* mice, which indicates RU-486 acute treatment had no effect on blood glucose. The lower blood glucose level in both groups may be the result of poor intake of food and water during recovery.

Similarly, in the second study (RU-486 + spironolactone), the body weight was found to be significantly higher in *db/db* compared with *db/+* in both treatment (35.74 ± 0. 7 vs. 23.06 ± 0.20 gm) and non-treatment (33.76 ± 0.35 vs. 23.10 ± 0.63 gm) groups at baseline, and remained significant at all time points. One-week treatment of RU-486 + spironolactone did not show any effect on body weight either in non-diabetic *db/+* or diabetic *db/db* mice compared with their vehicle control (as shown in Figure 2A). As expected, the blood glucose values were significantly higher in *db/db* mice compared with *db/+* in both treatment (473.13 ± 27.68 vs. 165.50 ± 12.16 mg/dl) and non-treatment (374.27 ± 68.69 vs. 178.50 ± 12.23 mg/dl) group at baseline as shown in Figure 2B. We did not see a change in blood glucose values after a week of treatment compared with vehicle-treated *db/db* mice (529.13 ± 32.54 vs. 567.25 ± 17.63 mg/dl). However, the values were significantly lower in *db/db*-treated group compared with *db/db* vehicle-treated group (501.10 ± 49.91 vs. 339.40 ± 64.59 mg/dl) 48 h post-HI. The result indicated that combined receptor blockade lessened the hyperglycemia without any change in body weight.

# Effect of RU-486 on body weight, blood glucose and stroke grading at 24 h post HI

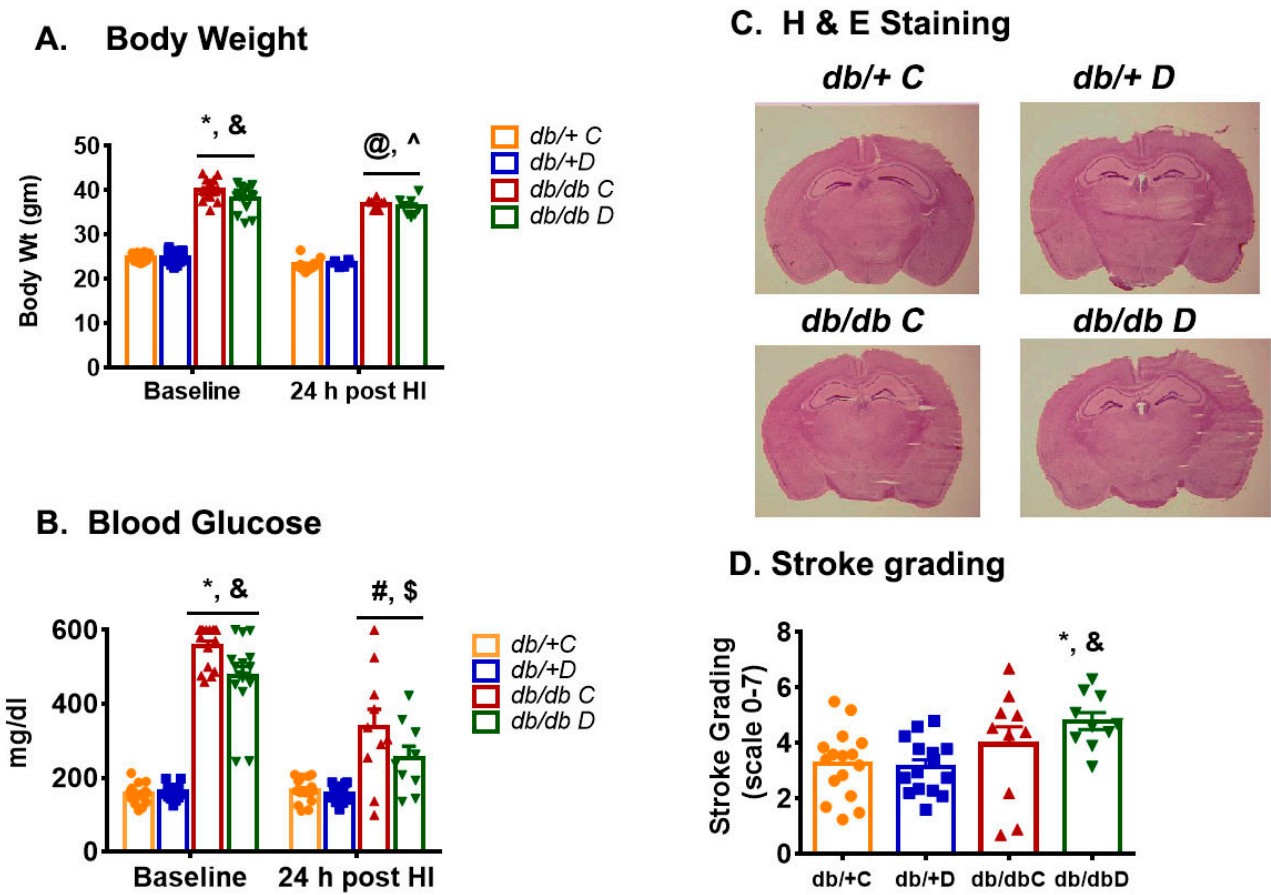

**Figure 1.** (**A**,**B**) shows the effect of RU-486 treatment on body weight and blood glucose in type-2 diabetic *db/db* and non-diabetic *db/+* at baseline and 24 h post-HI compared with their vehicle control. (**C**) shows the representative image of H & E staining at hippocampal level and (**D**) shows the stroke grading following RU-486 treatment on infarct size in type-2 diabetic *db/db* and non-diabetic *db/+*. Values were shown as mean ± SEM and significance were set at *p* < 0.05. ***db/+C:*** vehicle control (*n* = 8 to 16), ***db/+D*** RU-486 treatment (*n* = 6 to 15), ***db/db* C:** vehicle control (*n* = 10 to16), ***db/db* D:** RU-486 treatment (*n* = 6 to 10). * *p* < 0.05 vs. *db/+C* baseline: and & *p* < 0.05 vs. *db/+D* baseline: @ *p* < 0.05 vs. *db/+ C* post-HI: ˆ *p* < 0.05 vs. *db/+D* post-HI: # *p* < 0.05 vs. *db/db C*-baseline: $ *p* < 0.05 vs. *db/db D* baseline.

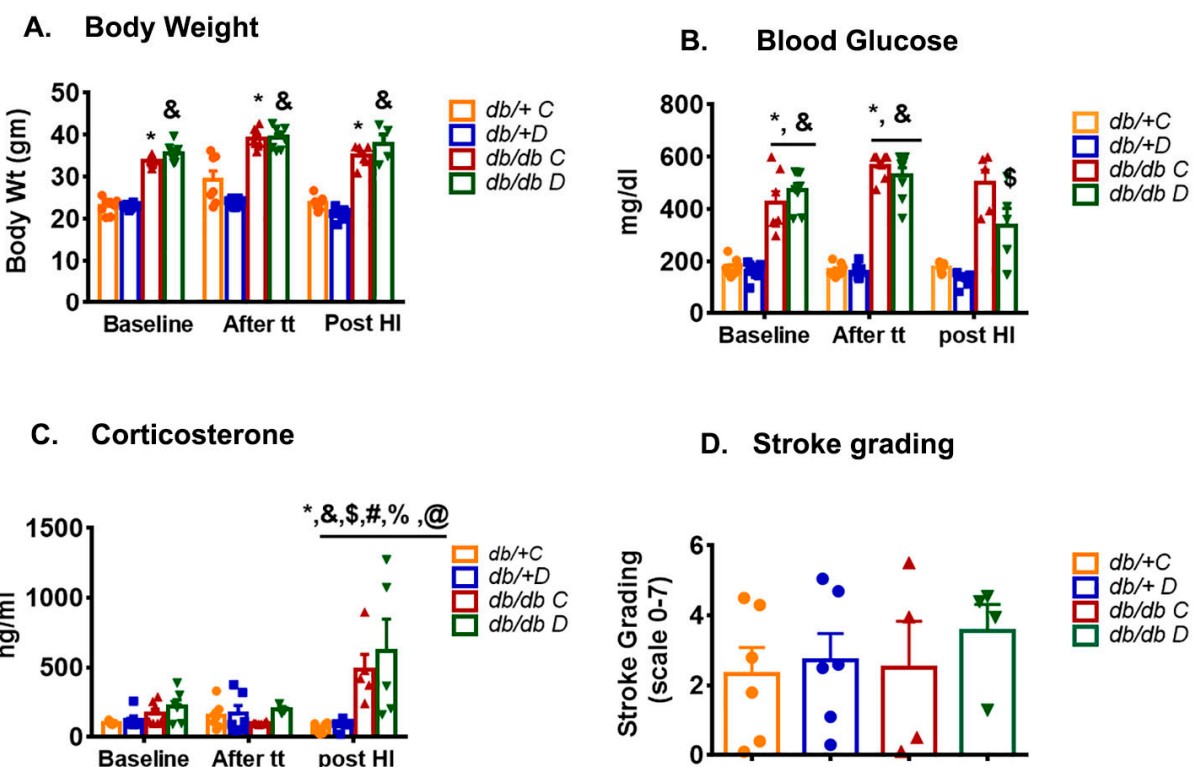

**Figure 2.** (**A**–**C**) shows the effect of RU-486 + spironolactone treatment on body weight, blood glucose and corticosterone at baseline, one week after treatment (tt) and 48 h post-HI compared with their vehicle control. Values were shown as mean ± SEM and significance were set at $p < 0.05$. ***db/+C:*** vehicle control (*n* = 5 to 8), ***db/+D***: RU-486 + spironolactone treatment (*n* = 5 to 8), ***db/db* C:** vehicle control (*n* = 5 to 8), ***db/db* D:** RU-486 + spironolactone treatment (*n* = 3 to 8), ***db/db* C-tt:** after 1week vehicle treatment, ***db/db* D-tt:** after 1 week of drug treatment. * $p < 0.05$ vs. *db*/+C baseline: and & $p < 0.05$ vs. *db*/+D baseline, # $p < 0.05$ vs. *db/db* C baseline: $ $p < 0.05$ vs. *db/db* D baseline, % $p < 0.05$ vs. *db/db* C -tt: @ $p < 0.05$ vs. *db/db* D-tt. (**D**) shows the effect of RU-486 + spironolactone treatment on infarct size (stroke grading), at 48 h post-HI compared with their vehicle control. Values were not significant.

*3.2. Effect of Glucocorticoid Receptor Blockers on Corticosterone*

Previously we observed a higher baseline corticosterone level in *ob/ob* mice compared with non-diabetic *ob/+* [20]. Similarly, in this RU-486 treatment study, we reported a higher level of corticosterone in *db/db* mice compared with *db/+* at baseline (311.40 ± 57.33 vs. 126.80 ± 21.54 ng/mL), and the release of corticosterone increased after hypoxia–ischemia in both *db/+* and *db/db* group compared with their baseline [27]. However, the significant increase in corticosterone was only observed in the RU-486-treated *db/+* not in the *db/db* mice when compared with their vehicle control at 24 h post-HI as shown earlier (Kumari et al., 2011).

In this study, again we observed a higher baseline corticosterone level in diabetic mice compared with non-diabetic control *db/+* (170.0 ± 30.88 vs. 103.24 ± 7.43 ng/mL). However, no significant change in corticosterone level was observed after a week of treatment of RU-486+ spironolactone either in *db/+*(123.93 ± 20.76 vs. 190.66 ± 20.76 ng/mL) or *db/db* (99.74 ± 6.43 vs. 199.43 ± 20.99 ng/mL) compared with their vehicle control. A significant increase in corticosterone was observed at 48 h post-HI in *db/db* mice compared with their baseline and after treatment (tt) groups as shown in Figure 2C. However, at 48 h post-HI

the difference between treatment and non-treatment was not found to be significant in *db/db* mice.

### *3.3. Effect of Glucocorticoid Receptor Blockers on Infarct Size*

We assumed that the glucocorticoid receptor blockers would normalize the hyperglycemia and lower the corticosterone and consequently reduce the infarct size. In this study, first, we pretreated the mice with GCRII blocker, RU-486 (40 mg/kg) 24 h before inducing stroke and during recovery. At 24 h post-HI, the results showed a significant increase in infarct size in *db/db* mice compared with *db/+*. However, no significant change in infarct size was noted after treatment either in non-diabetic *db/+* or *db/db* mice compared with their vehicle group as shown in Figure 1C (H & E staining) and Figure 1D (stroke grading).

We then tested the effect of both glucocorticoid receptor blockers I and II on infarct volume following a week of treatment before inducing stroke. We graded the stroke in different regions of the brain and found no significant difference between treatment and non-treatment group either in control *db/+* or *db/db* mice, as shown in Figure 2D.

### 4. Discussion

In this study, we determined the effects of GCR II blocker and combination of GCR I + II blocker on acute hyperglycemia following stroke and the measured the outcome in terms of infarct size. The results suggested that both GCR blockers, RU-486 and a combined treatment of RU-486 + spironolactone tend to reduce the hyperglycemia in type-2 *db/db* mice, and the effect was significantly different when treated in combination. However, the reduced hyperglycemia had no positive outcome in reducing infarct size.

It has long been appreciated that hyperglycemia is mediated by glucose-induced elevated plasma corticosterone. In an earlier study, despite successfully reducing corticosterone levels using metyrapone (11β-HSD1, synthesis inhibitor), hyperglycemia-induced brain damage was not prevented in a rat MCAO model [28]. In this study, we prevented the binding of corticosterone to the glucocorticoid receptor by a GCR blockers to reduce the function rather than preventing the synthesis of corticosterone. Therefore, we noted increased serum corticosterone in both RU-486 and RU-486 + spironolactone-treated mice following HI. In our initial study, we successfully reduced the corticosterone using metyrapone in a stress-induced model in C57BL6 mouse. Metyrapone is an inhibitor of enzyme11β- Hydroxysteroid dehydrogenase type-1, which converts inactive 11-dehydroxy corticosterone into active corticosterone which amplifies the glucocorticoid receptor mediated action. However, the treatment effect was transient due to a very short half-life of metyrapone (around 1.9 h), and multiple injections were required during post-stroke recovery, which may have added additional stress during stroke recovery (data not published). Therefore, in this study, we chose to block the receptor using RU-486, GCR II blocker, which has a slow metabolism and the plasma concentration of drug decreases to half by 12 to 72 h. Similarly, spironolactone has active metabolites until 13 to 16 h, therefore a single injection was suitable for this study. In the first study, we attempted to block the GCR II receptor in type-2 *db/db* mice and noted the decrease in blood glucose and increase in corticosterone level following RU-486 treatment. Similar to our findings, Liu et al. suggested that RU-486 treatment reversed the increases in glucocorticoid receptor expression and the enzyme responsible for corticosterone synthesis within liver, which reduced the blood glucose level, increased plasma corticosterone without any significant change in body weight and insulin level in type-2 diabetic mice [24]. In this study, mice received lower doses of RU-486, 25 mg/kg twice daily for 2–3 weeks, compared with a higher dose 40 mg/kg every 12 h in our study. The results showed a decrease in blood glucose in *db/db* mice that did not reach significance, compared with their vehicle control, however, this could be due to long-term vs. short term treatment of RU-486.

It has been shown that basal levels of GC only activates GCR1 receptor or mineralocorticoids, whereas inflammatory or stress related release of GCs activate both GCR1

and GCRII [29]. Therefore, in this study, we treated the diabetic mice with GCR1 blocker, spironolactone (25 mg/kg) and GCRII blocker, RU-486 (25 mg/kg) for one-week prior inducing stroke. The treatment strategy reduced hyperglycemia and increased the corticosterone level as reported previously [24] without any change in infarct size.

Previously, Soulet and Rivest et al. showed that RU-486 pretreatment decreased the survival rate of mice exposed to an intracerebral infusion of lipopolysaccharide (LPS) and suggested that blockade of glucocorticoid action at the receptor level increased synthesis of putrescine, which is responsible for the severe immune reaction in the CNS [30]. Similarly, another study of pretreatment of RU-486 in rat, caused a severe immune reaction, which was highly toxic for the cerebral tissue [31]. In this study we did not observe a difference in mortality between vehicle and RU-486-treated *db/db* mice at 24 h post-HI. However, we observed overall higher mortality in *db/db* compared with *db/+* in the RU-486 treatment group (37.5% vs. 6.25%) following post-HI. In the RU-486 + spironolactone study, mortality in *db/db* mice was lower in the treatment group compared with vehicle (25% vs. 50%) at 48 h post-HI. In fact, the overall mortality was similar in *db/+* and *db/db* treatment group. We do not know the exact reason for such differences in survival/mortality between RU-486 alone and RU-486 + spironolactone group. As mentioned above, an impaired HPA axis in the type-2 diabetic rodent may be a factor for the higher mortality rate. The beneficial or detrimental effects of an immune reaction in the brain is dependent upon the ability of glucocorticoids to provide the proper feedback mechanism. Once the level of corticosterone rises in the plasma, a feedback action activated on HPA axis and in this study, a compromised HPA axis in *db/db* mice may be the reason of poor outcome. Another possibility may be the lack of release of heat shock proteins (HSPs), which maintains the glucocorticoid and mineralocorticoid receptor in the inactive form [29]. It is possible that in this study, blocking of the GC receptors may have reduced the release of HSPs and consequently imbalance the action of GCs causing the adverse response.

In conclusion, this study suggested that pretreatment of glucocorticoid receptor II blocker and a combined treatment of glucocorticoid receptor I + II, lessened the hyperglycemic effect but provided no improvement in infarct size post-HI. The combination treatment showed improvement in the overall survival in diabetic mice. Future studies are warranted to understand the relationship, and the effects of glucocorticoid receptor blockers on hyperglycemia.

**Author Contributions:** R.K.: Conceptualization, data analysis, and manuscript preparation. L.W.: Animal study, histology, editing of the manuscript. All authors have read and agreed to the published version of the manuscript.

**Funding:** The study was supported by National Institute of Health, Funding number: DK75130 to Ian A Simpson.

**Institutional Review Board Statement:** All the experimental procedures in this study were approved by the Institutional Animal Care and Use Committee, College of Medicine, Penn State University under protocol number: # 2006028.

**Informed Consent Statement:** Not applicable.

**Data Availability Statement:** Data will be available upon request.

**Acknowledgments:** I would like to thank Ian A Simpson for conceptualization, supervision and providing resources for this study.

**Conflicts of Interest:** The authors declare no conflict of interest.

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
