# Peer review of "Glucocorticoid Receptor Blockers Pretreatment Did Not Improve Infarct Volume in Type-2 Diabetic Mouse Model of Stroke"

_diabetology, doi:10.3390/diabetology3040041_

Round 1

Reviewer 1 Report

In this paper, the authors tried to explore the effect of glucocorticoid receptor blockers in type-2 diabetic mouse model of stroke. The topic is of interest to readers, however, I do have some major concerns/ recommendations listed below regarding the manuscript.

1.       Title: it was not informative or conclusive. The authors should tell readers what the effect is with use of glucocorticoid receptor blockers in type-2 diabetic mouse model of stroke (e.g. beneficial and detrimental).

2.       Introduction: the authors did not describe intensively about the gap between what is known (the relationship between glucocorticoid and elevated blood glucose) and what is unknown in the field (the effect of glucocorticoid on stroke in the background of diabetes). Why this study is essential and important?

3.       Method:

1)      There were differences in the designs of drugs’ administration (RU-486 vs RU-486 & spironolactone): the time points and the frequencies. The definition regarding groups dividing was not clear: baseline, after tt and post HI.

2)      Controls missing: the controls should also include spironolactone only group, that is RU-486 vs spironolactone vs RU-486 & spironolactone.

4.       Results:

1)      Description was missing regarding symbols (&, # and $) in the legend of Figure 1.

2)      Figure 1A: the authors did not find a significant effect of RU-486 on blood glucose between treatment vs non-treatment at 24 h post HI in db/db mice, which indicate RU-486 had no inhibition in glucose levels in db/db mice?

3)      Figure 1B: the db/db C group had a big range of variation regarding stroke grading. What would contribute to the variation?

4)      Why the blood glucose and corticosterone levels were assessed at 24h post HI with treatment of RU-486 only while at 48h post HI with treatment of RU-486 & spironolactone?

5)      The authors mentioned that there was no difference in mortality between vehicle and RU- 486 treated db/db mice at 24 h post HI (line 244). How about the difference in survival/ mortality between vehicle and RU- 486 & spironolactone treated db/db mice at 48h post HI. I wonder whether the significantly decreased glucose values in the RU- 486 & spironolactone treated group (compared to db/db vehicle control) were caused by drug toxicity and poor intake of food and water during HI recovery, as there was no change in blood glucose levels after a week of treatment with RU- 486 & spironolactone compared to vehicle treated mice (Figure 2B).

5.       Conclusion: as commented above, to be cautious, it would require more solid evidence to conclude that pretreatment of glucocorticoid receptor II blocker and a combined treatment of glucocorticoid receptor I+II, lessened the hyperglycemic effect.

Author Response

Thanks to the reviewers for their comments and suggestions to improve the manuscript. Reviewer # 1 1. Title: Title has been changed as suggested. 2. Introduction: The background information related to glucocorticoids, stroke and diabetes has been updated in the introduction. 3. Method: 1) The study design has been inserted in the Methods section for better clarity of the groups. 2) We appreciate the reviewer’s suggestion regarding the spironolactone control group. We will consider that in future studies, however at this point, we are not able to provide that data. 4. Result: 1) Fig 1 legend has been updated. 2) The effect of RU-486 on blood glucose has been included in the results section. 3) Previous studies of ours have shown that variation in neuronal cell death in different brain regions is dependent upon the angiogenesis of the individual mouse brain, which would also be the reason for the range of stroke grading in this study as well. 4) In the initial study, we wanted to assess the acute effect of RU-486 on blood glucose and corticosterone due to the fact that major changes in the immune response and cytokines has been observed within 24 h post stroke. However, in the second study (RU-486+ Spironolactone treatment), mice were pretreated for 1 week prior to surgery, and we anticipated a protective effect of pretreatment on infarct size. Therefore, we analyzed the samples at 48 h post HI because the infarct develops fully around 48 h to72 h. 5) We did not observe a difference in mortality between vehicle or RU-486 treated db/db mice at 24 post HI. However, the mortality in RU-486 + spironolactone treated db/db mice was lower compared to vehicle group at 48 h post HI (25% vs 50%). In general, we observe 20 to 30% mortality in db/db mice at 48 h. We have included this information in the discussion section 6. We appreciate the suggestion about the effect of treatment on blood glucose in RU- 486 + spironolactone treated mice. In general, we do see the decreased body weight and blood glucose simultaneously in mice post HI. However, in this study the body weight was maintained after 48 h post HI in the treatment group, similar to the baseline and only a change in blood glucose was observed. The treatment effect may be mediated through glucocorticoid receptors because it is only evident following post stroke (or following stress) not after a week post treatment (without stress). 5. Conclusion: Conclusion has been modified as suggested.

Reviewer 2 Report

- The manuscript is interesting and the authors are attempting to prove a rather bold hypothesis where they are trying to see if blocking glucocorticoid receptors in a diabetic muse model would provide a better prognosis outcome for stroke. The experimental design and data are of good quality.

- The authors have used tried and tested glucocorticoid receptor blockers although they do not demonstrate the proof of concept that the receptors were indeed blocked. This would add value to the manuscript and prove that the effects they see were due to the blockers when compared to that of the vehicle.

- Figures should be rearranged to provide a better flow to the manuscript. Results keep jumping between RU486 and RU486+spironolactone.

- Figure presentation is also confusing and can be presented with more clarity. Groups that are compared must be paired together and depicted only. Rest of the data could go into supplementary information. 

Author Response

Thanks to the reviewers for their comments and suggestions to improve the manuscript. Reviewer # 2 - We appreciate the reviewer’s suggestion to include a receptor blocking study. However, in this study, we did not examine receptor blocking confirmation because they are FDA approved drugs, and the relationship between dose and receptor blocking activity has been extensively studied (Fleseriu M et al, 2011). In the animal study, 25-40 mg/kg dose has been used to effectively block the glucocorticoid receptors (Mracsko E et al, 2014, Liu Y et al, 2005). Therefore, in this study, we chose a safer and effective dose of RU-486 40 mg/kg dose for the acute study and lower dose, 25 mg/kg, for a longer treatment. - Now we have arranged the text in the result section for a better presentation and clarity.

Reviewer 3 Report

The manuscript by Rashmi Kumari et al., entitled "Effect of glucocorticoid receptor blockers in type-2 diabetic 2 mouse model of stroke" lacks significance, findings are mostly negative, and is poorly written. Authors must make the following changes before being considered for publication.

1. A schematic diagram explaining the study design is needed for readers to understand it easily and clearly. Methods should be more detailed eg how many times was the RU-486 injected, dose? As diurnal rhythm affects corticosterone levels, please specify the time of day blood was collected to measure corticosterone levels. 

2. Data needs to be presented as scatter dot plots with bar +SEM (eg Fig 1B) to represent all the data points.

3. Figure legends lack titles and there is no flow/connection when the results were written. eg results section ( 3.1) is titled Effect of glucocorticoid receptor blockers on body weight and blood glucose. However, the authors did not provide body weight data in Figure 1 and instead provided that in Figure 2. 

4. Authors should also provide H&E images used to quantify the infarct scores. 

Author Response

Thanks to the reviewers for their comments and suggestions to improve the manuscript. Reviewer # 3 1. Schematic diagram has been inserted as a study design in method sections for better understanding for the readers. The dose and the timing of drug administration and blood collection time is mentioned in the method sections. 2. All the data has been changed to scatter plot as suggested. 3. Body weight data is now included in Fig. 1 as per suggestion. 4. The representative image of H& E staining image has been included in Fig.1

Round 2

Reviewer 1 Report

The requested points are properly processed. I have no further queries.

Reviewer 3 Report

The authors have answered all of my questions/concerns and made the suggested corrections.